# Motivating factors, barriers and facilitators of participation in COVID-19 clinical research: A cross-sectional survey of Canadian community intensive care units

Jennifer L. Y. Tsang[1,2,3]*, Robert Fowler[4,5], Deborah J. Cook[6,7], Karen E. A. Burns[4,6,8], Kylee Hunter[9], Victoria Forcina[2], Anna Hwang[2], Erick Duan[1,3,7], Lisa Patterson[3], Alexandra Binnie[10]

1 Department of Medicine, McMaster University, Hamilton, Ontario, Canada, 2 Niagara Regional Campus, Michael G. DeGroote School of Medicine, McMaster University, St. Catharines, Ontario, Canada, 3 Niagara Health, St. Catharines, Ontario, Canada, 4 Interdepartmental Division of Critical Care, Temerty School of Medicine, Toronto, University of Toronto, Toronto, Canada, 5 Sunnybrook Health Sciences Centre, Toronto, Ontario, Canada, 6 Department of Health Research Methods, Evidence & Impact, McMaster University, Hamilton, Ontario, Canada, 7 St. Joseph's HealthCare, Hamilton, Ontario, Canada, 8 St. Michael's Hospital, Toronto, Ontario, Canada, 9 Faculty of Medicine, University of Ottawa, Ottawa, Ontario, Canada, 10 William Osler Health System, Etobicoke, Ontario, Canada

* Jennifer.tsang@mcmaster.ca

**Data Availability Statement:** All relevant data are within the manuscript and its supporting information files.

## Abstract

Only a small proportion of COVID-19 patients in Canada have been recruited into clinical research studies. One reason is that few community intensive care units (ICUs) in Canada participate in research. The objective of this study was to examine the motivating factors, barriers and facilitators to research participation amongst Canadian community ICU stakeholders. A cross-sectional online survey was distributed between May and November 2020. The survey focused on 6 domains: participant demographics, ICU characteristics, ICU research infrastructure, motivating factors, perceived barriers, and perceived facilitators. Responses were received from 73 community ICU stakeholders, representing 18 ICUs. 7/18 ICUs had a clinical research program. Participants rated their interest in pandemic research at a mean of 5.2 (Standard Deviation [SD] = 1.9) on a 7-point Likert scale from 'not interested' to 'very interested'. The strongest motivating factor for research participation was the belief that research improves clinical care and outcomes. The most significant facilitators of research involvement were the availability of an experienced research coordinator and dedicated external funding to cover start-up costs, while the most significant barriers to research involvement were a lack of start-up funding for a research coordinator and a lack of ICU research experience. Canadian Community ICU stakeholders are interested in participating in pandemic research but lack basic infrastructure, research personnel, research experience and start-up funding. Evolution of a research support model at community hospitals, where most patients receive acute care, may increase research participation and improve the generalizability of funded research in Canada.

**Funding:** JT received the Canadian Critical Care Trials Group Early Career Research Award to support this study. https://www.ccctg.ca/Home The funders had no role in study design, data collection and analysis, decision to publish, or preparation of the manuscript.

**Competing interests:** The authors have declared that no competing interests exist.

## Introduction

In the 20 months since the emergence of Coronavirus Disease-2019 (COVID-19), close to 210 million cases have been confirmed worldwide. Globally, more than 4.3 million deaths have been documented. As of February 10, 2022, Canada has recorded more than 3.16 million cases, resulting in more than 35,000 deaths [1].

Massive international efforts are underway to identify effective treatments for COVID-19 and several successful therapies have been identified including corticosteroids [2, 3] and IL-6 inhibitors [4]. However, the morbidity and mortality of COVID-19 remain high, particularly with the proliferation of new variants [5].

Although Canadian hospitals have contributed sizably to COVID-19 research investigations, only a small proportion of eligible COVID-19 patients in Canada have been enrolled in clinical trials. One of the reasons for Canada's low research participation is that relatively few Canadian hospitals participate in clinical research. This is particularly true of community hospitals, accounting for over two-thirds of Canadian hospital beds [6], which have historically had limited research involvement [7].

Community hospitals represent a large untapped resource for research in Canada. Patient recruitment in community hospitals would increase study enrolment and accelerate the pace of knowledge acquisition. It would also potentially engage a more diverse population of Canadian patients in research studies, including Canadians living in rural areas, the North, near-North, and more indigenous Canadians. Finally, in the context of the COVID-19 pandemic, it would ensure that research studies are implemented across many regions, recognizing that COVID-19 is most active in different regions at different times.

Recognizing the need for broader participation in COVID-19 pandemic research, we designed, tested and administered a web-based cross-sectional survey of Canadian community ICUs to understand the motivating factors, facilitators and barriers to participation in COVID-19 pandemic research, with the goal of developing strategies to increase research engagement in community hospitals.

## Materials and methods

### Survey development and testing

We conducted this survey from May 2020 to November 2020. To generate relevant survey items, two investigators (J.T and A.B) reviewed the literature for motivating factors, barriers and facilitators of research participation. Six investigators (J.T., A.B., K.E.B, R.F, D.J.C., and E. D) generated and grouped the survey items into 6 domains, including participant demographics, hospital research infrastructure, ICU characteristics, motivating factors, perceived facilitators, and perceived barriers. The final survey included multiple question formats (yes/no, nominal, ordinal and Likert scales and open-ended questions). (S1 Appendix).

Participants were also asked to evaluate 3 models of financial support for community ICU participation in pandemic research: in Model A, the ICU would receive a partial subsidy for a research coordinator (RC) salary and retain full autonomy over the research program, including receiving all study payments. In Model B, an individual study sponsor would hire an RC externally who would work remotely recruiting patients for that study only, and all study payments would revert to the sponsor. The ICU would be free to join whichever studies they preferred but would not receive study payments. Finally, in Model C, an external sponsor would hire an RC to recruit locally for a package of studies. All study payments would revert to the sponsor and the ICU would have limited choices in terms of which studies to join. In addition to indicating their preferred model, participants were asked to explain their rationale for their preferred model. (See S1 Appendix for additional details.)

The pilot questionnaire was reviewed by three critical care physicians for relevance and flow.

## Survey administration

The target population was healthcare professionals working in Canadian community ICUs. Eligible survey respondents included any ICU staff member including critical care physicians, nurses, allied health professionals (pharmacists, physiotherapists, respiratory therapists, occupational therapists, registered dietitians), research staff (research coordinators, research assistants, research managers) or hospital administrators. The survey was disseminated through the Canadian Critical Care Trials Group, the Canadian Community ICU Research Network, the Canadian Critical Care Society by word of mouth, email and through social media platforms associated with these groups. We sent a single reminder through the Canadian Community ICU Research Network. Due to the self-selected and non-probabilistic nature of the sample, invitations and response rates could not be quantified.

## Statistical analysis

Data are described as mean and standard deviation or median and interquartile range (IQR) for continuous variables and number and percentage for categorical variables; data from Likert scales were treated as continuous variables. Continuous variables were compared using the parametric Student's t-test or the non-parametric Kruskal-Wallis or Wilcoxon rank sum tests.

## Ethical consideration

Ethics approval was obtained from the Hamilton Integrated Research Ethics Board (HiREB #11101). Informed consent was obtained electronically via online survey tool (REDCap).

# Results

## Participant demographics

We received responses from 73 Canadian community ICU professionals, representing 18 community ICUs. Most participants worked in Ontario (78.1%). Female respondents outnumbered male respondents (60.3 vs. 39.7%). The largest professional group was physicians (32.9%), followed by registered nurses (30.1%), pharmacists (8.2%), hospital administrators (8.2%), nurse practitioners (6.8%), respiratory therapists (6.8%), and research coordinators (5.5%) Fig 1.

## Characteristics of ICUs

The 18 ICUs represented by respondents ranged in size from fewer than 10 ventilated beds to over 40 ventilated beds, with a median of 16–20 beds. All of the ICUs cared for medical patients (18/18) while the majority also cared for surgical patients (17/18). Care for specialized patient populations such as coronary care (7/18), neurosurgical care (1/18) and trauma (1/18) were uncommon while none of the ICUs reported providing care for burns (0/18), cardiovascular surgery (0/18) or transplant (0/18) populations. During the first wave of the COVID-19 pandemic (February to June 2020), the peak number of COVID-19 patients in each ICU varied significantly from 1–5 patients (9/18) to 6–10 patients (2/18), 11–15 patients (4/18) and 16–20 patients (3/18).

## Pre-existing research infrastructure in community ICUs

Participants were asked about the availability of research infrastructure in their hospitals and ICUs. Responses revealed that a majority of institutions had research policies and procedures

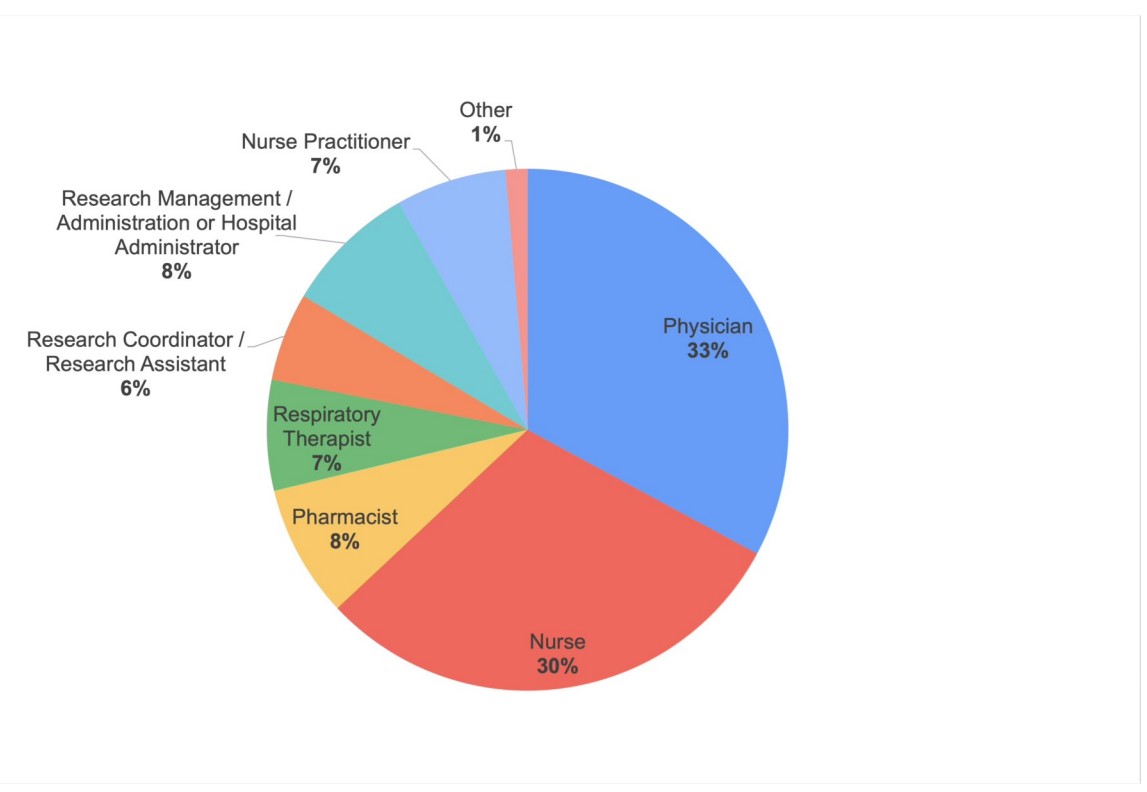

**Fig 1. Participant demographics–by professions.**

(10/18) as well as a research ethics board (9 local, 4 remote). A majority also had a pre-existing research program within the hospital (10/18), although not necessarily in the ICU, and a pharmacy department with research capability or experience (9/18). Few than half of the community ICUs had pre-existing ICU research programs (7/18) or access to an ICU research coordinator or assistant (6/18). In addition, only a minority of ICUs reported a hospital research administration office (7/18), on-site contract review capability (4/18), a clinical laboratory with research capability or experience (2/18) and a diagnostic imaging department with research capability or experience (2/18) Table 1.

To gauge the degree of local research expertise, participants were asked about their personal research experience: 64% had experience caring for patients enrolled in clinical research studies, 45% had research experience as a trainee, 15% had experience as a principal investigator, 21% as a local site investigator or co-investigator, and 18% as a research coordinator or research assistant. In addition, 23% had research training at the graduate level, 27% through online research courses, and 30% had basic science research experience. In total, 86% of participants had at least one form of research experience. Amongst physicians, 92% had at least one form of research experience and 54% had experience as a principal investigator, local site investigator or co-investigator Fig 2.

## Participant research interest and motivating factors for COVID-19 research

Respondents were asked about their interest in participating in COVID-19 clinical research on a 7-point Likert scale from 'not interested (1)' to 'very interested (7)'. The mean level of interest was 5.2 (SD = 1.9), with 81% of participants reporting an interest level of 4 or above and 55%

**Table 1. Pre-existing hospital research infrastructure (N = 18).**

| Pre-existing Research Infrastructure | Percentage of Community Hospitals (%) |
|---|---|
| Pre-existing ICU research program | 39 |
| Pre-existing research program(s) in other clinical department(s) in the hospital | 56 |
| Research coordinator(s) or research assistant (s) in the ICU | 33 |
| Research coordinator(s) or research assistant (s) in other clinical department (s) in the hospital | 33 |
| Local research ethics board | 50 |
| Remote research ethics board | 22 |
| Hospital research administration/office | 39 |
| On-site contract review capability | 22 |
| Research policies and procedures | 56 |
| Pharmacy department with research capability or experience | 50 |
| Clinical laboratory department with research capability or experience | 11 |
| Diagnostic imaging department with research capability or experience | 11 |
| Other | 11 |

reporting an interest level of 6 or 7 Fig 3. Physicians reported greater interest in COVID-19 research (mean = 5.8, SD = 1.5, n = 24) than did allied health professionals, including pharmacists, respiratory therapists, speech pathologists, and administrators (mean = 4.4, SD = 2.1, n = 18) (p = 0.017). Research coordinators reported the highest level of interest in COVID-19 research (mean = 7.0, SD = 0, n = 4). Nurses were intermediate in their interest level (mean = 5.0, SD = 2.1, n = 27) and were not significantly different from physicians.

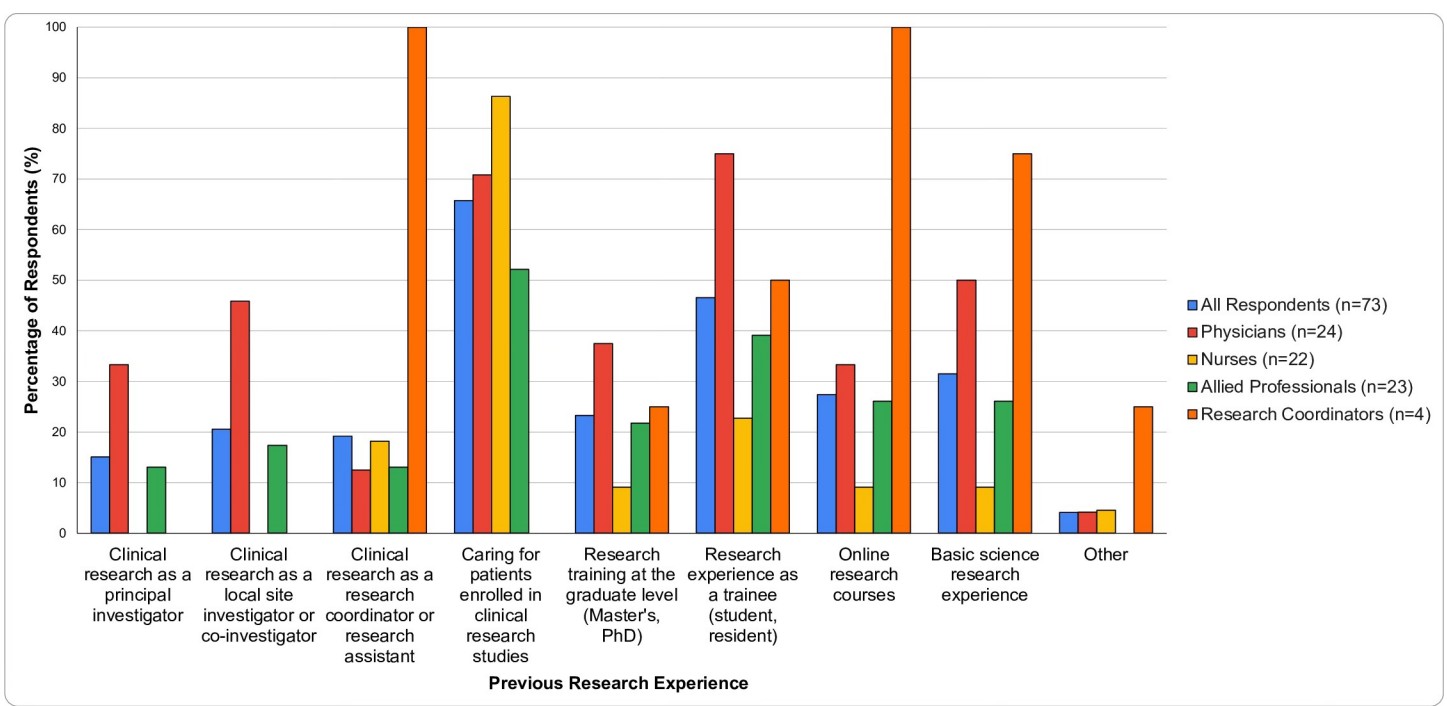

**Fig 2. Participants previous research experience (N = 73).**

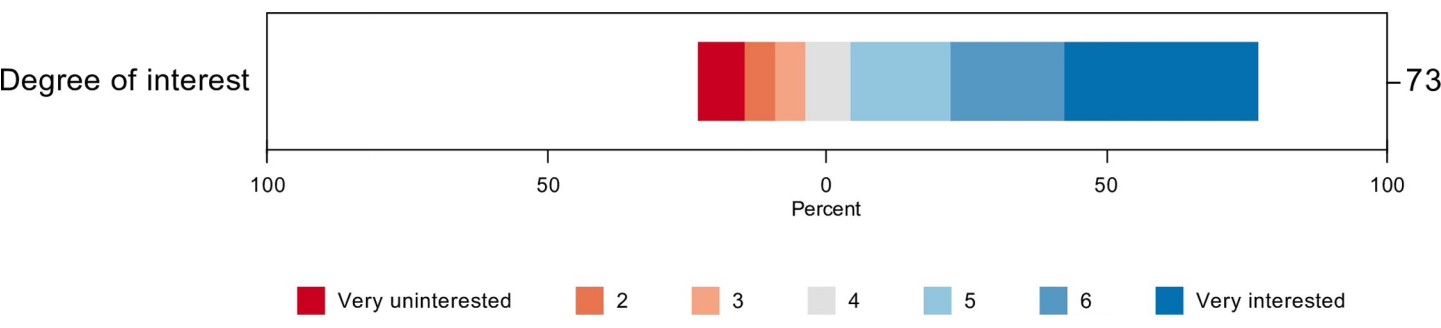

**Fig 3. Level of research interest by professions.**

Participants were asked to rate 12 potential motivating factors for research participation on a 5-point Likert scale ranging from 'strongly disagree (1)' to 'strongly agree (5)'. The top 5 motivating factors were: research improves clinical care and outcomes (4.7, SD = 0.6), research advances medical knowledge (4.6, SD = 0.5), participation in research allows participants to stay informed about current research (4.3, SD = 0.7), participation in research allows participants to establish collaborations with other ICUs/ICU research (4.2, SD = 0.8), and research enhances professional opportunities in the hospitals (4, SD = 0.9) Fig 4.

## Barriers to community ICU research participation

Respondents were also asked to evaluate potential barriers to community ICU research involvement on a 5-point Likert scale from 'not significant (1)' to 'very significant (5)'. The 5 most significant barriers were: lack of start-up funding for research coordinator (3.8, SD = 1.1), inadequate per-patient payments to sustain research coordinator salary (3.6, SD = 1.0), staff workload is too high due to pandemic-related pressures (3.5, SD = 1.2),

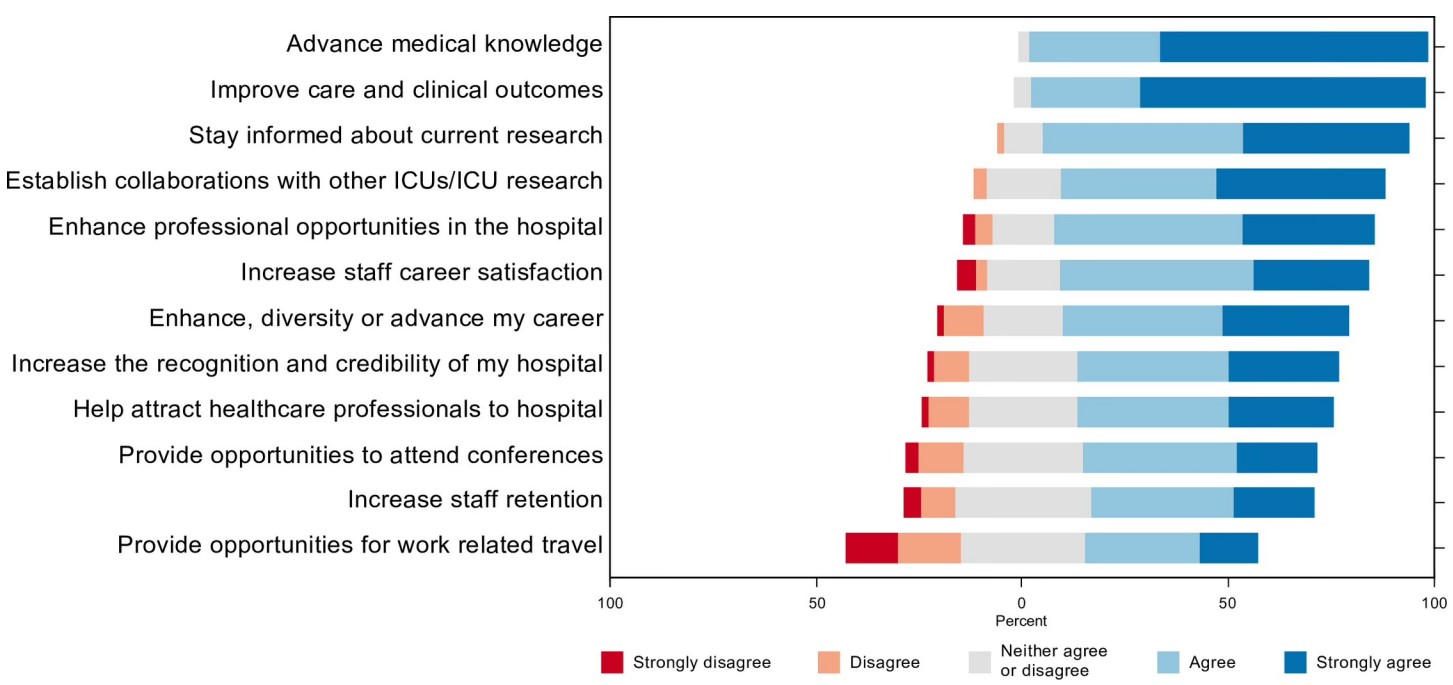

**Fig 4. Motivating factors for research.**

community ICUs are not known or expected to do research (3.4, SD = 1.2), and lack of research experience in ICU (3.3, SD = 1.2) Fig 5.

Respondents also proposed the following additional barriers to COVID-19 research participation: inability of healthcare professionals to hold dual clinical and research jobs due to hospital regulations, lack of patients to be recruited to many clinical trials, unwillingness of hospital administrators to reassign modified-duty healthcare professionals to take on research roles, and lack of compensation for healthcare professionals for the additional work related to research.

## Facilitators of community ICU research participation

Respondents were asked to evaluate potential facilitators of community ICU research involvement on a 5-point Likert scale from 'not significant (1)' to 'very significant (5)'. The top 5 most significant facilitators were: dedicated external funding to provide start-up costs for research program (4.3, SD = 1.0), dedicated external funding to sustain ongoing costs for research program (4.3, SD = 0.9), availability of an experienced research coordinator (4.2, SD = 1.1), participation in research networks that support community hospitals (4, SD = 1.2), and partnership with academic hospitals (3.9, SD = 1.3) Fig 6.

Respondents also proposed the following additional facilitators: increased per-patient enrolment payment to improve research program sustainability, simplified research protocols and data collection to reduce cost, creation of regional research programs to provide logistical support, and provision of research funding from global hospital budgets.

## Financial support models for community ICUs to participate in COVID-19 pandemic research

Respondents were asked to indicate their preferred models of financial support for community ICU participation in COVID-19 clinical research, based on 3 proposed models (see Methods

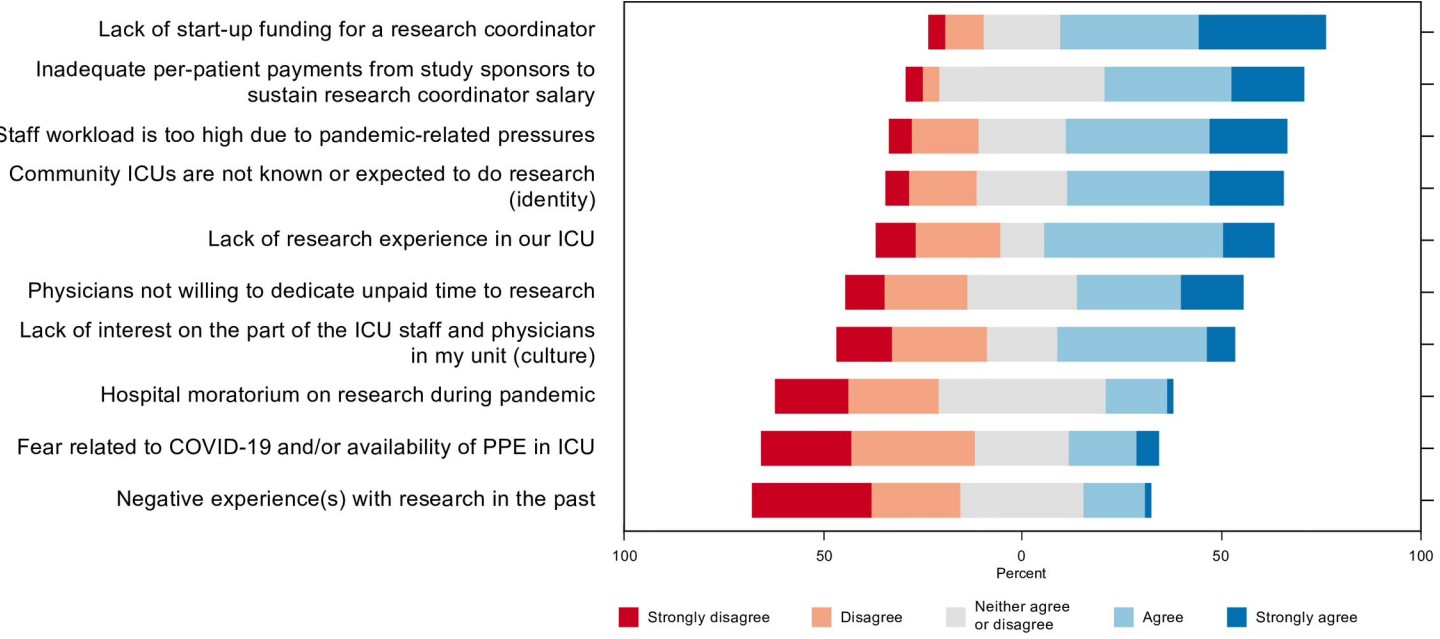

**Fig 5. Barriers to community ICU research participation.**

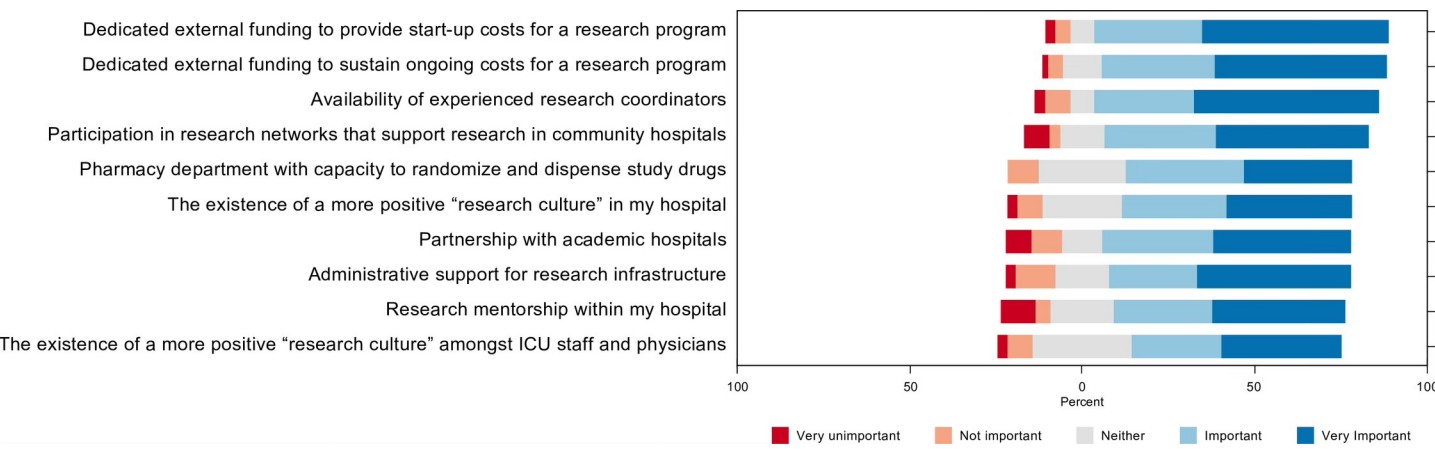

**Fig 6. Facilitators of community ICU research participation.**

section). Most preferred Model A (56.5%), in which the ICU would receive a partial subsidy from study sponsor to assist with the costs of hiring a research coordinator. Participants felt that this model retained autonomy while allowing their ICU to build local research capacity. Model B, in which research coordinators would be hired by individual studies to recruit patients remotely for a particular study, was preferred by 15.9% of participants. These respondents stated that Model B was more sustainable for smaller hospitals and would have less impact (time and financial commitment) on the ICU as a whole. Finally, Model C, in which an external sponsor would hire and pay for a coordinator to work locally for a package of studies in the ICU with all payments reverting to the sponsor was preferred by 27.5% of participants. These respondents considered that Model C was more flexible and offered the highest level of financial support.

## Discussion

To our knowledge, this is the first study exploring the level of interest, motivating factors, barriers and facilitators of community ICU participation in COVID-19 pandemic research. Our results demonstrate that interest in COVID-19 research is high, particularly amongst physicians and research coordinators. Most stakeholders believe that engaging in research would result in better patient care and advances in medical knowledge, while also improving job satisfaction.

These beliefs are consistent with studies suggesting that participation in research enhances patient care. Snihur *et al* reported a growing body of evidence suggesting that hospitals participating in research have better patient outcomes for coronary artery disease, colorectal cancer and ovarian cancer [8–12]. The benefits were primarily driven by the formation of innovative, research-oriented medical teams, enhanced training opportunities for hospital staff, rapid adoption of new clinical guidelines and the attraction and retention of talent [13].

However, our study also reveals many perceived barriers to research participation in community hospitals, including a lack of financial support and the absence of experienced research coordinators to initiate and implement research programs. When asked about models of external financial support, most respondents preferred a model in which a partial subsidy was provided to help cover research coordinator salary, thereby reducing the financial risk associated with the participation in clinical research.

Our results highlight a much-needed discussion about the one-size-fits-all approach to research funding in Canada. Clinical research studies typically pay an umbrella of start-up fee

to cover start-up costs, followed by a per-patient fee for each patient enrolled. However, these payments presume that there is a pre-existing research program at the participating centre. Start-up costs for centres without pre-existing research infrastructure can be significant and are not factored into study payments. These costs may include establishing research infrastructure (research ethics boards, research administration, contract review), creating research policies and procedures, hiring and training research coordinators, training pharmacy staff, nurse managers and physicians in research best practices, and negotiating research protocols with hospital departments such as laboratory, pharmacy, and diagnostic imaging. In the absence of funding to support these start-up costs, the challenges of creating a new research program from the ground up may be insurmountable.

In addition to a lack of financial support, a lack of research experience was also identified as a barrier to clinical research participation. One solution is to partner community hospitals with academic hospitals in a research mentoring relationship. For example, St. Joseph's Healthcare, an academic hospital in Hamilton, Ontario, has established a close research partnership with the ICUs at Niagara Health, a community hospital network in the Niagara Region, and at Brantford General Hospital in Brantford, Ontario. These partnerships have enabled both Niagara Health and Brantford General Hospital to start-up and sustain robust clinical research programs incorporating many trials. An alternative mentorship model exists in Alberta where academic and community hospitals are geographically clustered into 'Health Zones'. In the Edmonton Health Zone, all three of the community ICUs: Sturgeon Community Hospital, Grey Nuns Community Hospital, and Misericordia Hospital, have been actively participating in the CATCO trial of COVID-19 therapies [14].

Finally, developing a national research network to link community hospitals that are interested in conducting research could empower community hospitals to participate in clinical trials [15]. The Canadian Cancer Clinical Trial Network is an example of a network that links community hospitals to each other as well as to nearby academic hospitals to facilitate research program development and clinical trial enrolment [16].

## Limitations

Our study has several limitations. Survey respondents were self-selected and responses may not be generalizable to all Canadian community ICU stakeholders. However, respondents were recruited from both small and large community ICUs, as well as those with and without ICU research programs, to capture a variety of perspectives. Respondents from Ontario were also overrepresented relative to those from other provinces, however, Ontario has a high proportion of community ICUs and several of the largest community hospitals and hospital networks in Canada are located in Ontario.

## Conculsion

Canadian Community ICU stakeholders are interested in participating in pandemic research and recognize the importance of clinical research for improving clinical care and outcomes. However, there is a lack of basic research infrastructure in community ICUs, including research personnel, research experience, and start-up funding. Evolution of a research support model at community hospitals, where most patients receive their acute care, may increase research participation and improve the generalizability of funded research in Canada.

## Supporting information

**S1 Data.**
(XLSX)

**S1 Appendix.**
(PDF)

## Acknowledgments

Jennifer L.Y. Tsang holds a McMaster University Department of Medicine Mid-Career Research Award. Deborah J. Cook holds a Canada Research Chair from the Canadian Institutes of Health Research. Karen E.A. Burns holds a PSI-50 Mid-Career Clinical Research Award. Rob Fowler is the H. Barrie Fairley Professor of Critical Care at the University Health Network & Interdepartmental Division of Critical Care Medicine.

## Author Contributions

**Conceptualization:** Jennifer L. Y. Tsang, Robert Fowler, Deborah J. Cook, Karen E. A. Burns, Kylee Hunter, Victoria Forcina, Erick Duan, Lisa Patterson, Alexandra Binnie.

**Formal analysis:** Jennifer L. Y. Tsang, Kylee Hunter, Victoria Forcina, Anna Hwang, Lisa Patterson, Alexandra Binnie.

**Funding acquisition:** Jennifer L. Y. Tsang.

**Investigation:** Jennifer L. Y. Tsang, Alexandra Binnie.

**Methodology:** Jennifer L. Y. Tsang, Alexandra Binnie.

**Project administration:** Jennifer L. Y. Tsang.

**Resources:** Jennifer L. Y. Tsang.

**Supervision:** Jennifer L. Y. Tsang.

**Validation:** Jennifer L. Y. Tsang, Alexandra Binnie.

**Writing – original draft:** Jennifer L. Y. Tsang.

**Writing – review & editing:** Jennifer L. Y. Tsang, Robert Fowler, Deborah J. Cook, Karen E. A. Burns, Kylee Hunter, Victoria Forcina, Anna Hwang, Erick Duan, Lisa Patterson, Alexandra Binnie.

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
