## [Decision Letter · Decision Letter 0]

9 Feb 2022

PONE-D-21-33770Motivating Factors, Barriers and Facilitators of Participation in COVID-19 Clinical Research A Cross-Sectional Survey of Canadian Community Intensive Care UnitsPLOS ONE

Dear Authors,

Thank you for submitting your manuscript to PLOS ONE. After careful consideration, we feel that it has merit but does not fully meet PLOS ONE’s publication criteria as it currently stands. Therefore, we invite you to submit a revised version of the manuscript that addresses the points raised during the review process.

We look forward to receiving your revised manuscript.

Kind regards,

Gebisa Guyasa Kabito, MPH

Academic Editor

PLOS ONE

Journal Requirements:

3. We note that you have stated that you will provide repository information for your data at acceptance. Should your manuscript be accepted for publication, we will hold it until you provide the relevant accession numbers or DOIs necessary to access your data. If you wish to make changes to your Data Availability statement, please describe these changes in your cover letter and we will update your Data Availability statement to reflect the information you provide

Reviewers' comments:

Reviewer's Responses to Questions

**Comments to the Author**

1. Is the manuscript technically sound, and do the data support the conclusions?

Reviewer #1: Yes

Reviewer #2: Yes

2. Has the statistical analysis been performed appropriately and rigorously? 

Reviewer #1: Yes

Reviewer #2: N/A

3. Have the authors made all data underlying the findings in their manuscript fully available?

Reviewer #1: Yes

Reviewer #2: Yes

4. Is the manuscript presented in an intelligible fashion and written in standard English?

Reviewer #1: Yes

Reviewer #2: Yes

5. Review Comments to the Author

Reviewer #1: For abstract: Include mean/SD/p value in results section

Conclusion for abstract

Cite line 73 (introduction)

The study was well conducted and written.

Hope it will give the merits of improving COVID 19 research in your community and elsewhere.

Reviewer #2: This is an online survey, and as such it gives useful information about what canadian intensivists think is needed to increase clinical research. Most of the critical factors identified as motivating, facilitating or obstructing, are probably of general validity.

6. PLOS authors have the option to publish the peer review history of their article (what does this mean?). If published, this will include your full peer review and any attached files.

Reviewer #1: No

Reviewer #2: No

---

## [Author Response · Author response to Decision Letter 0]

19 Mar 2022

Dear PLOS One Editors and Reviewers,

Thank you for spending the time reviewing our manuscript despite your busy schedule and providing comments.

We have made changes to our manuscript as suggested by editors and reviewers. Please see our responses in red below.

We have also made some minor editorial changes to the rest of the manuscript. These changes are marked via tracked changes. I have included both the tracked changes file and clean file. 

Reviewer #1: For abstract: Include mean/SD/p value in results section Mean and SD included. P value is not applicable here.

Conclusion for abstract I am unclear on what is requested here.

Cite line 73 (introduction) Completed 

Thank you for the comment. We have made the above changes accordingly. 

The study was well conducted and written.

Hope it will give the merits of improving COVID 19 research in your community and elsewhere. 

Thank you.

Reviewer #2: This is an online survey, and as such it gives useful information about what Canadian intensivists think is needed to increase clinical research. Most of the critical factors identified as motivating, facilitating or obstructing, are probably of general validity. 

Thank you.

1. Please ensure that your manuscript meets PLOS ONE's style requirements, including those for file naming. Completed. The PLOS ONE style templates can be found at 

https://journals.plos.org/plosone/s/file?id=wjVg/PLOSOne_formatting_sample_main_body.pdf and https://journals.plos.org/plosone/s/file?id=wjVg/PLOSOne_formatting_sample_main_body.pdf and https://journals.plos.org/plosone/s/file?id=ba62/PLOSOne_formatting_sample_title_authors_affiliations.pdf

Done.

2. Please provide additional details regarding participant consent. In the ethics statement in the Methods and online submission information, please ensure that you have specified (1) whether consent was informed and (2) what type you obtained (for instance, written or verbal, and if verbal, how it was documented and witnessed). If your study included minors, state whether you obtained consent from parents or guardians. If the need for consent was waived by the ethics committee, please include this information. Informed consent was obtained electronically via online survey tool (REDCap). We have amended the manuscript in ethics statement and online submission information.

If you are reporting a retrospective study of medical records or archived samples, please ensure that you have discussed whether all data were fully anonymized before you accessed them and/or whether the IRB or ethics committee waived the requirement for informed consent. If patients provided informed written consent to have data from their medical records used in research, please include this information. N/A

3. We note that you have stated that you will provide repository information for your data at acceptance. Should your manuscript be accepted for publication, we will hold it until you provide the relevant accession numbers or DOIs necessary to access your data. If you wish to make changes to your Data Availability statement, please describe these changes in your cover letter and we will update your Data Availability statement to reflect the information you provide Will do.

Thank you. We look forward to hearing from you.

Jennifer Tsang, MD, PhD, FRCPC, ABOM Diplomate

Research Lead, Intensivist, Niagara Health

Physician Co-Lead, Critical Care Research Program, Niagara Health

Regional Deputy Research Director, Internal Medicine Residency Program, McMaster University

Associate Professor of Medicine, McMaster University

March 6, 2022

---

## [Editor Report · Decision Letter 1]

28 Mar 2022

Motivating Factors, Barriers and Facilitators of Participation in COVID-19 Clinical Research A Cross-Sectional Survey of Canadian Community Intensive Care Units

PONE-D-21-33770R1

Dear Authors,

We’re pleased to inform you that your manuscript has been judged scientifically suitable for publication and will be formally accepted for publication once it meets all outstanding technical requirements.

Kind regards,

Gebisa Guyasa Kabito, MPH

Academic Editor

PLOS ONE
---

## [Editor Report · Acceptance letter]

6 Apr 2022

PONE-D-21-33770R1 

Motivating factors, barriers and facilitators of participation in COVID-19 clinical research: a cross-sectional survey of Canadian community intensive care units 

Dear Dr. Tsang:

I'm pleased to inform you that your manuscript has been deemed suitable for publication in PLOS ONE. Congratulations! Your manuscript is now with our production department. 

Kind regards, 

on behalf of

Dr. Gebisa Guyasa Kabito 

Academic Editor

PLOS ONE